# Genome-Wide Identification and Expression Analysis of *BBX* Transcription Factors in *Iris germanica* L.

**DOI:** 10.3390/ijms22168793

**Published:** 2021-08-17

**Authors:** Yinjie Wang, Yongxia Zhang, Qingquan Liu, Ting Zhang, Xinran Chong, Haiyan Yuan

**Affiliations:** 1Institute of Botany, Jiangsu Province and Chinese Academy of Sciences, Nanjing 210014, China; wangyinjie@cnbg.net (Y.W.); zhangyongxia@cnbg.net (Y.Z.); liuqingquan@cnbg.net (Q.L.); zhangting901014@cnbg.net (T.Z.); chongxr@cnbg.net (X.C.); 2The Jiangsu Provincial Platform for Conservation and Utilization of Agricultural Germplasm, Nanjing 210014, China

**Keywords:** *Iris germanica* L., *BBX* transcription factors, stress response, phytohormone, expression profiling

## Abstract

The family of B-box (BBX) transcription factors contains one or two B-BOX domains and sometimes also features a highly conserved CCT domain, which plays important roles in plant growth, development and stress response. Nevertheless, no systematic study of the *BBX* gene family in *Iris germanica* L. has been undertaken. In this study, a set of six *BBX* TF family genes from *I. germanica* was identified based on transcriptomic sequences, and clustered into three clades according to phylogenetic analysis. A transient expression analysis revealed that all six BBX proteins were localized in the nucleus. A yeast one-hybrid assay demonstrated that *IgBBX3* has transactivational activity, while *IgBBX1*, *IgBBX2*, *IgBBX4*, and *IgBBX5* have no transcriptional activation ability. The transcript abundance of *IgBBXs* in different tissues was divided into two major groups. The expression of *IgBBX1*, *IgBBX2*, *IgBBX3* and *IgBBX5* was higher in leaves, whereas *IgBBX4* and *IgBBX6* was higher in roots. The stress response patterns of six *IgBBX* were detected under phytohormone treatments and abiotic stresses. The results of this study lay the basis for further research on the functions of BBX gene family members in plant hormone and stress responses, which will promote their application in *I. germanica* breeding.

## 1. Introduction

Transcription factors (TFs) play primary roles in gene expression regulation. According to their conserved domains, TFs can be divided into different gene families. The B-box (BBX) proteins form a class of zinc finger transcription factors containing one or two B-box domains near their N-terminus, and some members also feature a highly conserved CCT (CONSTANS, CO-like, and TOC1) domain near their C-terminus [1]. In *Arabidopsis*, the BBX TF family has 32 members and divided into five groups according to the number of B-box domains and the presence of a CCT domain. Groups I (AtBBX1 through AtBBX6) and II (AtBBX7 through AtBBX13) have two B-boxes and a CCT domain, whereas group III (AtBBX14 through AtBBX17) has one B-box and a CCT domain; Group IV (AtBBX18 through AtBBX25) has two B-box domains without the CCT domain and Group V (AtBBX26 through AtBBX32) has only a single B-box domain [2,3]. The BBX TF family has 30 members in the rice (*Oryza sativa*) genome [4], 25 members in the pear (*Pyrus bretschneideri Rehd*) genome [5], 28 members in the petunia (*Petunia hybrid*) genome [3] and 64 members in the apple (*Malus domestica* Borkh.) genome [6]. However, to date, there have been no studies about the members of *BBX* family in *Iris germanica* L.

BBXs are key factors are involved in the regulation of growth and development, including flowering, shade-avoidance response, seedling photomorphogenesis, biotic and abiotic stresses and plant hormonal pathways [1,3,7,8]. *BBX* TFs are best known for regulating flowering. *AtBBX1*/*CO* is a core factor in the flowering pathway under long day conditions [9]. *CO* regulates the expression of the Flowering Locus T (*FT*) gene, a prominent floral inducer [10]. The group I and II members *COL3* (*AtBBX4*), *COL5* (*AtBBX6*) and *COL9* (*AtBBX7*) were reported to be involved in flowering regulation [11,12,13]. The group IV members *AtBBX19* and *AtBBX24* were also reported to regulate flowering [14,15]. Overexpression of the *BBX* group V gene *EIP6* (*AtBBX32*), has been shown to repress flowering under long day conditions [16].

Several studies have been reported that the BBX family members are also involved in abiotic stress responses and hormonal signaling networks. In *Arabidopsis thaliana*, *AtBBX18* plays a negative role in thermotolerance [17]. Overexpression of *BBX24* in *Arabidopsis* can enhance salinity tolerance [18]. Meanwhile, over-expression of *OsBBX25* has been reported to enhance the salt and drought tolerance of *A. thaliana* [19]. In *Chrysanthemum* (*Chrysanthemum morifolium*), *CmBBX24* has been shown to improve tolerance to drought and low temperature [20]. In *Solanum* plants, *SsBBX24* gene responded to PEG and salt treatments, but not to water or low temperature deficit [21]. BZS1, a B-box protein, negatively regulates the brassinosteroid and light signaling pathways [22].

*Iris germanica* L., which is often called Pogon Iris, is one of the most popular ornamental species in the genus *Iris* [23]. To our knowledge, little information has been reported on the isolation and functional analysis of *BBX* TFs in *I. germanica*. Here, we isolated six *BBX* TFs in *I. germanica* based on a set of transcriptomic data. Subsequently, the transcription levels of *BBX* TFs in different tissues, and under various stress and phytohormone treatments, were investigated. The results provide a foundation for further functional characterization of *BBX* genes in *I. germanica*.

## 2. Results

### 2.1. Identification and Phylogenetic Analysis of IgBBX Genes in *I. germanica*

The six isolated *BBX* sequences were designated as *IgBBX1* through *IgBBX6* (GenBank: MW357644-MW357649). Full-length cDNA varied from 749 to 1478 bp, and their predicted protein products comprised from 197 (IgBBX3) to 475 (IgBBX6) residues. Full details of the *IgBBX* sequences are given in Table 1. The conserved domains of IgBBX were identified by the SMART and Pfam database (Appendix A). Among these six IgBBX proteins, IgBBX1, IgBBX2 and IgBBX6 contained two B-BOX domains plus a conserved CCT domain. IgBBX3, IgBBX4, and IgBBX5 consisted of two B-BOX domains without a CCT domain. The protein sequence alignment and motifs logos showed that the B-Box 1 and B-Box 2 domains of the IgBBX had similar conserved amino acid residues, and the CCT domain was also highly conserved in the IgBBX family (Appendix A).

To evaluate the evolutionary relationship between *Arabidopsis* and *I. germanica* BBX proteins, the deduced amino acid sequences of the BBX genes identified were completely aligned. A combined phylogenetic tree was then constructed using the neighbor-joining method and bootstrap analysis (1000 reiterations) (Figure 1). IgBBX1 and 2 belong to BBX subfamily group Ⅰ, while IgBBX3, 4 and 5 all belong to BBX subfamily group Ⅳ, and IgBBX6 belongs to BBX subfamily group Ⅱ.

### 2.2. Subcellular Localization and Transcription Activation of IgBBX Genes

To obtain evidence that the *IgBBXs* acted as transcription factors, the subcellular localization of six *IgBBXs* was investigated by transient expression in tobacco epidermal cells with a transgene comprising *IgBBX* fused to green fluorescent protein (GFP) driven by the CaMV35S promoter. The experiment showed that the *IgBBX*-GFP activity was restricted to the nucleus and colocalized with the nuclear marker D53-mCherry, while empty GFP was ubiquitously distributed in both the cytoplasm and the nucleus (Figure 2).

*IgBBXs* were further fused to the DNA binding domain (BD) to investigate the transcription activation activity in yeast cells. The negative control pGBKT7 or pGBKT7-*IgBBX3* construct was unable to grow on either the SD/-His-Ade containing X-α-gal medium or SD/-His-Ade medium, whereas the positive control (pCL1), or pGBKT7-*IgBBX1*, pGBKT7-*IgBBX2*, pGBKT7-*IgBBX4*, pGBKT7-*IgBBX5*, and pGBKT7-*IgBBX**6* constructs grew extremely well on both media (Figure 3).

### 2.3. Expression Profile of IgBBX in Different Plant Tissues

Since no *BBX* factors have been previously documented in *I. germanica*, we investigated the expression profiles of these genes. The results showed six *IgBBX* genes were differentially transcribed throughout the plant (Figure 4). The expression of *IgBBX1*, *IgBBX2*, *IgBBX3* and *IgBBX5* was higher in leaves, whereas *IgBBX4* and *IgBBX6* was higher in roots. The expression of *IgBBX2*, *IgBBX3* and *IgBBX5* was barely expressed in roots.

### 2.4. Expression Profile of IgBBX Genes after ABA, MeJA and SA Phytohormones Treatment

The transcription levels of *IgBBX1* and *IgBBX2* were substantially up-regulated after a 3 and 6 h of exposure to ABA; however, *IgBBX3*, *IgBBX4* and *IgBBX5* were down-regulated throughout the treatment, and *IgBBX6* was only marginally induced at 3 and 6 h (Figure 5a). The *BBX* family genes exhibited different expression patterns under MeJA treatment. *IgBBX1* and *IgBBX2* were induced by the treatment, whereas *IgBBX3* and *IgBBX5* were repressed. The transcripts of *IgBBX4* were increased at 3 h, but the transcript abundance then declined, whereas *IgBBX6* was only induced at 24 h (Figure 5b). When subjected to SA treatment, *IgBBX3* was induced at 3 h but was down-regulated thereafter. *IgBBX1* and *IgBBX2* were induced within 1 h, whereas the expression of *IgBBX4* and *IgBBX5* was down-regulated by the entire treatment and the expression of *IgBBX6* was increased at the end of the period (Figure 5c).

### 2.5. Transcription Profiling of IgBBX Genes under Abiotic Stress

Regarding the response to salinity stress, *IgBBX1* and *IgBBX4* were all up-regulated after a 3 h exposure to NaCl, whereas the expression of *IgBBX6* was strongly induced at both the 6 and 12 h time points. The transcript abundance of *IgBBX2* was down-regulated throughout the treatment. *IgBBX3* was strongly down-regulated after 6 h, while *IgBBX5* was repressed at 12 h but increased after 24 h (Figure 6a). When challenged with PEG6000, *IgBBX1* was induced at 12 h but repressed after 24 h. The expression of *IgBBX2* was significantly decreased at all time points. The expression of *IgBBX3*, *IgBBX4* and *IgBBX5* were gradually increased over the course of the first 6 h, after which they fell away somewhat, then gradually increased to a normal level at 24 h. *IgBBX6* was rapidly increased after 3 h of treatment and then decreased to the normal level at 24 h (Figure 6b). Under low temperature, *IgBBX1*, *IgBBX3*, *IgBBX4* and *IgBBX5* were all initially induced, but later moderately repressed. *IgBBX2* was rapidly down-regulated in contrast to the up-regulation of *IgBBX6* (Figure 6c). *IgBBX1* and *IgBBX4* were strongly up-regulated by wounding at 1 h, but later rapidly repressed at 3 h and then significantly increased after 6 h. *IgBBX2* and *IgBBX5* were up-regulated at 1 h, but later rapidly repressed after 3 h and, then gradually increased to normal level at 24 h. *IgBBX3* was rapidly repressed at 3 h, then gradually increased to normal level at 24 h. *IgBBX6* was strongly up-regulated by wounding at 6 h and 12 h, and then significantly decreased after 12 h (Figure 6d).

## 3. Discussion

BBX protein is one of the important transcription factors that plays an important role in many developmental processes including flowering, seedling photomorphogenesis, shade-avoidance response, responses to biotic and abiotic stresses and plant hormonal pathways [1]. The function and evolution of *BBX* genes have been identified in *Arabidopsis* [14], rice (*Oryza sativa* L. cv. *Nipponbare*) [24], *Chrysanthemum* [25], pear (*Pyrus pyrifolia*) [26], and tomato (*Solanum lycopersicum*) [27]. However, little is known about the *I. germanica* BBX family. In this study, six *BBX* genes were identified in *I. germanica* based on transcriptome data. The BBX family was divided into five classes according to their conserved domains in *Arabidopsis* [28]. To clarify the phylogenetic relationships among the *IgBBX* genes, a combined phylogenetic tree was constructed based on the alignment of *Arabidopsis* and *I. germanica* BBX. According to the phylogenetic tree, six *IgBBXs* were divided into three groups (Figure 2), which were similar to those of *Arabidopsis* and rice [4,28]. All six of the BBX family proteins were located, as would be expected for TFs, in the nucleus (Figure 2), and consistent with the previous results [26,29,30]. Most BBX proteins have been proven to act as transcriptional activators of downstream genes [2,31], and yeast-based transient transformation assays suggested that *IgBBX1*, *2*, *4*, *5* and *6* possessed trans-activational activity (Figure 3). It is therefore probable that in planta these five TFs function as transcriptional activators of various downstream genes.

Gene expression patterns can provide important clues for gene function, and qRT-PCR was used to examine the expression of *IgBBX* genes in roots, leaves and flowers (Figure 4). The expression profiles revealed that the expression of *IgBBX* in different tissues showed spatial variations. The expression of *IgBBX3* and *5* was widely expressed in leaves and flowers, indicating that they might play roles in regulating leaf and flower development, which is consistent with previously described functional roles of *BBX* genes during flower development [14]. The expression of *IgBBX4* and *6* was highly expressed in the roots, indicating that they possibly regulate the development of roots in *I. germanica*. However, additional studies are needed to determine the functions of these *IgBBX* genes.

As BBX family TFs play important roles in plant development and stress responses [3], we investigated the responses of *IgBBXs* to various plant hormones and abiotic stress treatments. The results showed that *IgBBXs* were both up-regulated and down-regulated by the treatments (Figure 5 and Figure 6), suggesting that *IgBBXs* might be involved in responses to various phytohormones that elicit a stress response. Emerging evidence suggests that pathways regulated by the phytohormones ABA, SA and MeJA are involved in a substantial amount of crosstalk between biotic and abiotic stress signaling pathways [32]. In chrysanthemum (*Chrysanthemum morifolium*), *CmBBX24* plays a dual role, modulating both flowering time and abiotic stress tolerance by influencing GA biosynthesis [20]. ABA treatments down-regulated the expression of *CmBBX19* in *Chrysanthemum morifolium*, and *CmBBX19* modulated drought tolerance mainly through inducing changes in the expression of ABA-dependent pathway genes, such as *CmRAB18* and *CmRD29B* [25]. qRT-PCR analysis showed that the expression of *MdBBX10* was significantly induced by exogenous abscisic acid (ABA) treatment in apple roots and leaves [33]. Here, *IgBBX3*, *IgBBX4* and *IgBBX5* were down-regulated by ABA application, whereas *IgBBX1*, *IgBBX2* and *IgBBX6* were not (Figure 5A), which was similar to results with apple (*Malus domestica* Borkh.) [6]. However, little is known about the role of the *BBX* gene family in SA and MeJA hormonal signaling pathways. Our results may provide the basis for advancing studies on *BBX* family genes in stress phytohormone signaling.

Accumulating evidence suggests that many BBX proteins are involved in the regulation of plant abiotic stress responses [27]. In this study, it was clear that the various stress treatments affected the transcription level of different combinations of 6 *IgBBX* genes, suggesting that most of them were involved in the stress response of *I. germanica*. In pear, 16 out of 37 *PbBBXs* were regulated by drought treatment, and 13 out of 16 were up-regulated or down-regulated within 12 h of dehydration [34]. *CmBBX19*, a homolog of *AtBBX19* is down-regulated by drought stress [25], similar to its homolog *IgBBX3*. For PEG stress, the expression of *CmBBX22*, an ortholog of *AtBBX22*, gradually increased in abundance over the course of the first 6 h, and then decreased slightly in *Chrysanthemum morifolium* [35]. Similarly, expression of *IgBBX4*, its *I. germanica* homolog, gradually increased over the course of the first 6 h, after which it somewhat decreased, and then gradually increased to the normal level at 24 h by PEG treatment, suggesting a conserved function in drought tolerance of *IgBBX4*. In *Petunia*, three *PhBBX* genes responded to drought stress, eight *PhBBX* genes responded to salt stress, and 18 *PhBBX* genes responded to cold stress [3]. In apple, the expression of *MdBBX10* was strongly induced by NaCl and polyethylene glycol in roots and leaves [33]. In grapevine, some stress-related cis-acting elements (low-temperature, drought and wound) were found in the promoter regions of the *VvBBX* genes [36]. Here, the expression of *IgBBX3* was up-regulated by salt, drought and wound stresses (Figure 6), suggesting *IgBBX3* might play an important role in response to multiple abiotic stress networks. Since very few studies have investigated the role of BBX genes in heat, cold and wound stresses, this work lays the basis for further research on the functions of *BBX* gene family members in different abiotic stress tolerance.

## 4. Materials and Methods

### 4.1. Plant Materials and Stress Treatments

*Iris germanica* L. cultivar (cv.) ‘2010200’ as the plant material was grown in the *Iris* Resource Collection Garden of Institute of Botany, Nanjing Sun Yat-Sen Memorial Botanical Garden (118°78′–119°14′E; 31°14′–32°37′N) P. R. China. Test plants were obtained from clonally propagated, 1-year-old plants were grown in a plant incubator (MT8070iE, shoreline Technology, Xubang, Jinan, China) under controlled conditions (22 °C, 40% relative humidity and 160 μmol m^−2^ s^−1^ light) and LD conditions (16: 8 h, light: Dark).

A variety of abiotic stresses were imposed, including 200 mM NaCl for salinity treatment and 20% *w*/*v* polyethylene glycol (PEG 6000) for drought treatment [37], low temperature (4 °C) and wounding. For NaCl and PEG 6000 assays, young plants were transferred to liquid medium containing the stress agent, and the leaves were sampled at various time points [38]. Cold stress was imposed at 4 °C in a plant incubator (Haier, Qingdao, China). The wounding treatment involved cutting the second fully expanded leaf. The phytohormone treatments involved spraying the leaves with either 50 μM abscisic acid (ABA) [39], 1 mM methyl JA (MeJA) [40] or 200 μM SA [41]. Leaves were separately collected prior to the stress treatment and then after 0, 1, 3, 6, 12, and 24 h of each treatment, and all the material was immediately frozen in liquid nitrogen and stored at −80 °C. Each treatment was replicated three times.

### 4.2. Isolation and Sequencing of Full-Length IgBBX cDNAs

Total RNA was extracted from leaves using the RNAiso reagent (TaKaRa, Tokyo, Japan) following the manufacturer’s protocol. The first cDNA strand was synthesized from a 1 μg aliquot of total RNA using SuperScript III reverse transcriptase (Invitrogen, Carlsbad, CA, USA). All of the putative *BBX* sequences downloaded from TAIR (https://www.arabidopsis.org/ accessed on 15 April 2021) were used to query the *Iris lactea* var. *chinensis* transcriptome data [42,43]. Based on the transcriptome database of *I**. lactea* var. *chinensis* [42,43], primer pairs (Appendix A) were designed to amplify a fragment of the *IgBBX* sequences, and RACE PCR was then used to obtain the full-length cDNA. For the 3′-RACE, the oligo (dT) primer dT-AP was used to synthesize the first-strand cDNA, followed by a nested PCR using the adaptor primer (dT-R) (Appendix A). For the 5′-RACE, the AAP and AUAP primers provided with 5′-RACE System kit v2.0 (Invitrogen, Carlsbad, CA, USA) were used in a nested PCR (Appendix A). PCR products were purified using the Biospin Gel Extraction kit (Bio Flux, Hangzhou, China) and cloned into the pMD19-T easy (TaKaRa, Tokyo, Japan) for sequencing. Finally, pairs of gene-specific primers (IgBBXx-ORF-F/R, Appendix A) were designed to amplify each open reading frame (ORF) sequence. The conserved motif of IgBBX proteins were analyzed by the Multiple Em for Motif Elicitation website (MEME, https://wwwmeme.sdsc.edu/meme/meme.html accessed on 29 July 2021). Subsequently, the IgBBX proteins were further verified for the presence of the B-BOX domain using SMART (http://smart.embl-heidelberg.de/ accessed on 29 July 2021) and Pfam (http://pfam.sanger.ac.uk/ accessed on 29 July 2021) searches.

### 4.3. Sequence Alignment and Phylogenetic Tree Construction

*A. thaliana BBX* sequences were downloaded from the *Arabidopsis thaliana* transcription factor database [44] and combined with the newly acquired *IgBBX* sequences to perform a multiple alignment analysis based on ClustalW software [45]. The subsequent phylogenetic tree was derived using MEGA version 7.0 software (https://www.megasoftware.net/home accessed on 8 May 2021) [46], applying the Neighbor-Joining algorithm with 1000 bootstrap replicates. The MEME v4.10.2 program [47] served to identify the motifs present in the six IgBBX proteins using the parameter settings suggested by Huang et al. [19], and retaining only motifs associated with an *E* value < 1 × 10^−5^.

### 4.4. Subcellular Localization of IgBBX Genes

The *p35S::GFP-IgBBX* constructs were introduced into tobacco epidermal cells via the Agrobacterium-infiltrated tobacco (*Nicotiana benthamiana*) leaf method [26]. The plasmid of *IgBBX* for transient transformation was generated using the Invitrogen Gateway system, according to the manufacturer’s instructions. The *IgBBX* ORFs, lacking the stop codon, were amplified using a Phusion^®^High-Fidelity PCR kit (New England Biolabs, Ipswich, MA, USA) (primers listed in Appendix A), then inserted into the pMD19-T easy vector (TaKaRa, Tokyo, Japan) to allow its sequencing-based validation. Each *pENTR^TM^1A*-*IgBBX* plasmid was previously subjected to the LR Clonase™ II enzyme mix (Invitrogen, Carlsbad, CA, USA) reaction to obtain GFP-fused constructs using the binary vector pMDC43, resulting in the plasmid *p35S::GFP-IgBBX*. For transient expression, *Agrobacterium tumefaciens* strain EHA105 carrying the pMDC43 and pMDC43-*IgBBX* was grown separately to OD600 = 0.8 and coinfiltrated with the p19 strain into leaves of five-week-old *N. benthamiana*. The YFP fluorescent signals were monitored 48 to 72 h after infiltration using a confocal laser scanning microscopy (LSM 700, Carl Zeiss).

### 4.5. Transactivation Activity Assay of IgBBX Genes

The ORF (open reading frame) of each *IgBBX* was cloned into the pGBKT7 vector (Clontech, Mountain View, CA, USA) at the EcoR I/BamH I or EcoR I/Pst I restriction sites (primer sequences given in Appendix A) to generate the *pBD-IgBBX* construct, which was then transformed into the yeast strain Y2H (Clontech, Mountain View, CA, USA) [48]. Selection for transformants carrying either one of the *pBD*-*IgBBXs,* or an empty pGBKT7, was selected by culturing on SD/-Trp medium, while the pCL1 transformants were selected on SD/-Leu medium. All three classes of transformant cells were transferred to an SD/-His-Ade medium supplemented with 20 mg/mL or 0 mg/mL X-α-gal to observe cell growth. Since the expression of *His3* is regulated by the GAL4-BD region, a *IgBBX* TF possessing activation ability should bind to the GAL4-BD upstream promoter sequence of *His3*, thereby activating its expression and enabling the transformed cells to grow on SD/-His-Ade containing 20 mg/mL X-α-gal [48].

### 4.6. Transcription Analysis of the IgBBX Genes

Transcription analysis was achieved using qRT-PCR, based on the IgBBX-RT-F/R primers listed in Appendix A. Each 20 μL amplification reaction mixture contained 10 μL SYBR^®^
*Premix Ex Taq™* II (TaKaRa, Tokyo, Japan), 0.4 μL of each primer (10 μM), 4.2 μL H2O and 5 μL cDNA template. The PCR cycling regime comprised an initial denaturing step of 95 °C for 2 min, followed by 40 cycles of 95 °C for 10 s, 55 °C for 15 s and 72 °C for 20 s, after which a melting curve analysis was conducted following each assay to confirm the specificity of the amplicons. The *UBC9* gene (GenBank: MT302552) was employed as a reference sequence [23]. Relative changes in each gene expression level were quantitated based on three biological replicates via the 2^−∆∆Ct^ method [49].

### 4.7. Data Analysis

The relative transcription levels of each *IgBBX* were log2 transformed and the profiles were compared using Cluster v3.0 software [50] and visualized using Treeview [51]. The data were analyzed by Student’s *t*-test using SPSS v17.0 software (SPSS Inc., Chicago, IL, USA).

## 5. Conclusions

Our study provided the first genome-wide analysis of the *BBX* gene family in *I. germanica*. The expression of six *IgBBX* TFs in response to various phytohormones and abiotic stresses treatments was characterized. The results of this study lay the basis for further research on the functions of the *BBX* gene family members in different abiotic stress tolerances, which will promote their application in *I. germanica* breeding.

## Figures and Tables

**Figure 1 ijms-22-08793-f001:**
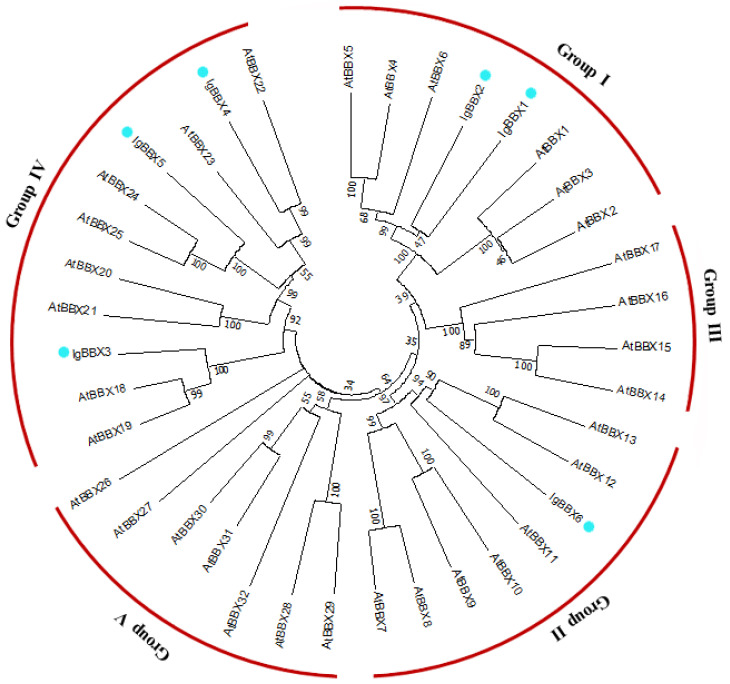
Phylogenetic analysis of BBX peptide sequences of *I. germanica* and *A. thaliana.* Sequences were aligned using ClustalW software and the subsequent phylogenetic tree constructed applying the Neighbor-joining algorithm. The red arcs indicate the various groups defined by the presence of known BBX domains. Dots indicate likely homologs.

**Figure 2 ijms-22-08793-f002:**
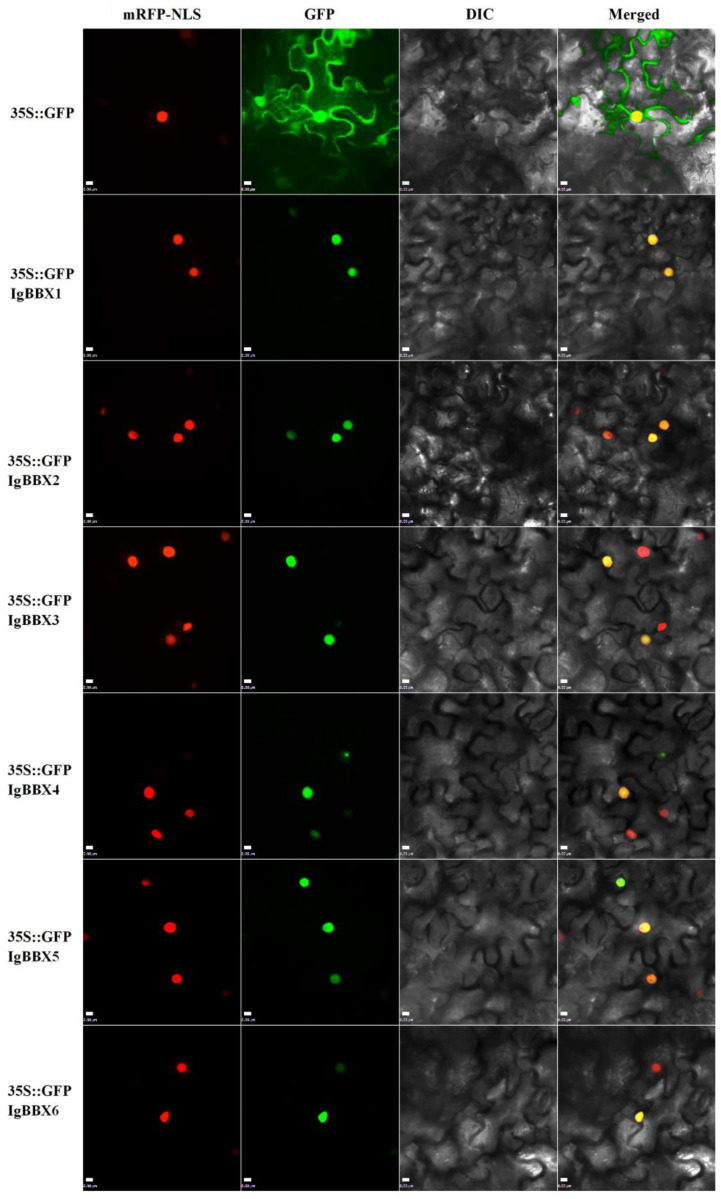
The subcellular localization of transiently expressed *IgBBX* TF fusion transgene in tobacco (*N. benthamiana*). The upper row shows the effect of the control p35S::GFP transgene and the lower rows that of the test transgene p35S::*GFP*-*IgBBX*. Bars = 50 μm. mRFP-NLS: a mRFP-labelled nuclear marker.

**Figure 3 ijms-22-08793-f003:**
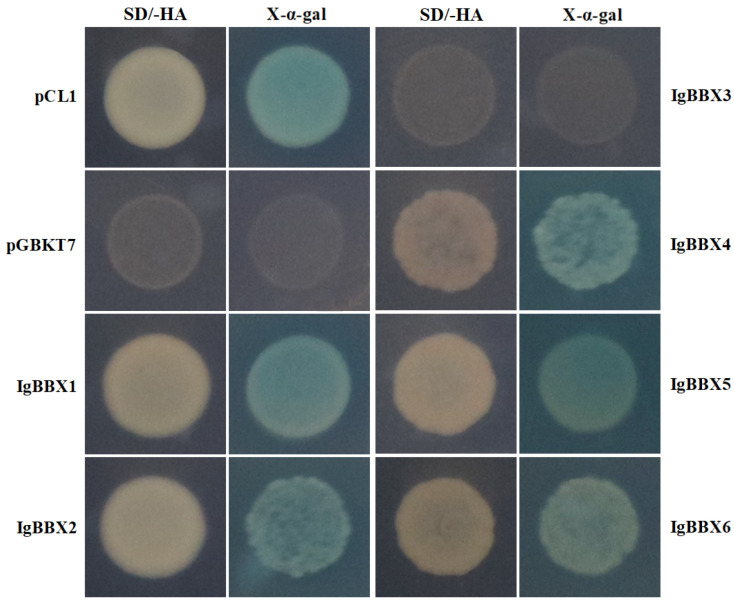
Transcriptional activity of the *IgBBX* TFs in yeast. SD/His-/Ade-: SD medium lacking histidine and adenine (on left side); X-a-gal: SD/His-/Ade- medium containing 20 mg/mL X-α-gal (on right side). pCL1 and an empty pGBKT7 are positive and negative controls, respectively.

**Figure 4 ijms-22-08793-f004:**
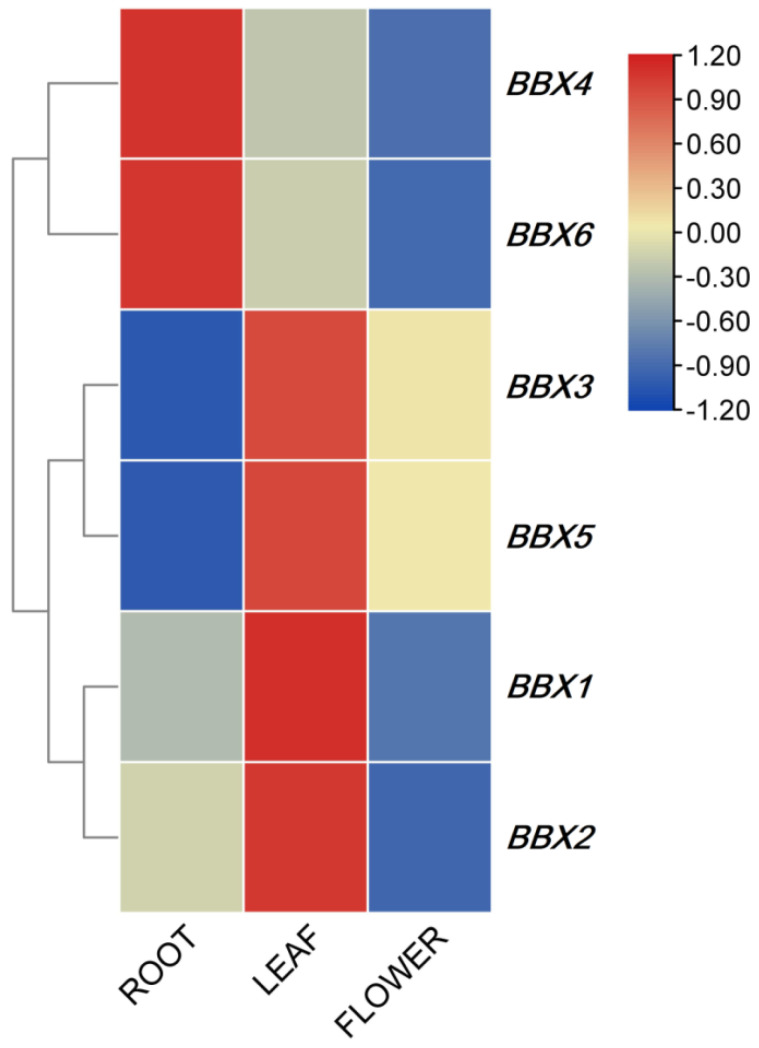
Expression profiles of *IgBBX* genes in various tissues. Blue and red indicate down-regulated and up-regulated transcript abundance, respectively, compared to the relevant controls (root). Bar on the top right corner represents log 2 transformed values.

**Figure 5 ijms-22-08793-f005:**
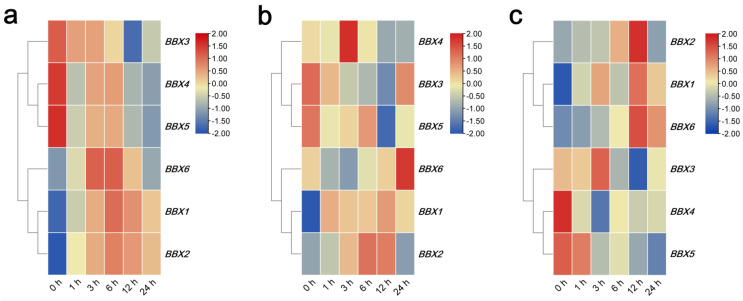
Expression profiles of *IgBBX* genes in leaf tissue following spraying with (**a**) abscisic acid (ABA), (**b**) methyl jasmonate (MeJA), and (**c**) salicylic acid (SA) treatments. Blue and red indicate down-regulated and up-regulated transcript abundance, respectively, compared to the relevant controls (0 h sample is an untreated sample). Bar on the top right corner represents log 2 transformed values.

**Figure 6 ijms-22-08793-f006:**
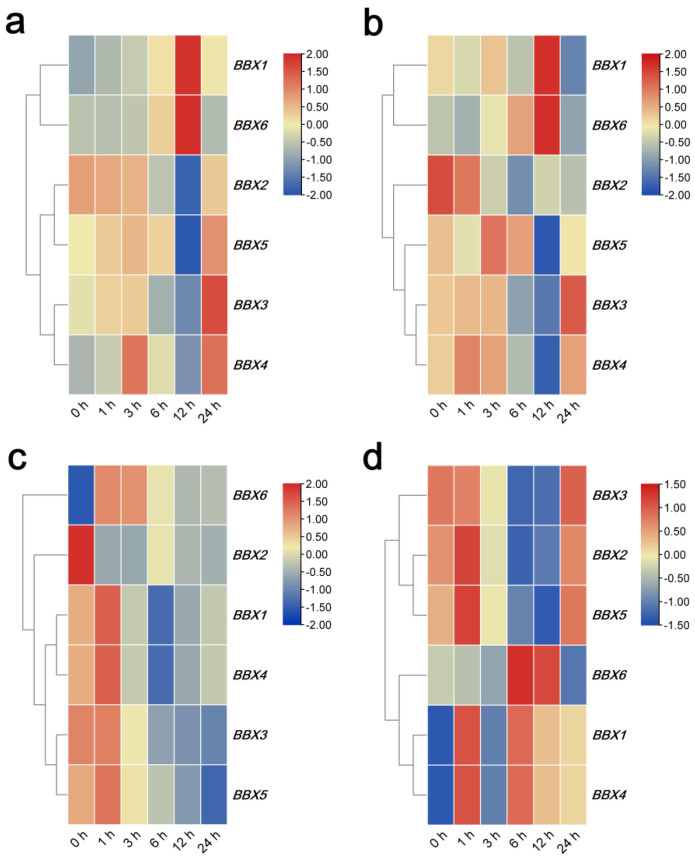
Expression profiles of *IgBBX* genes in leaf tissue following exposure to (**a**) salinity stress, (**b**) drought stress, (**c**) high temperature (40 °C), and (**d**) wound treatments. Blue and red indicate down-regulated and up-regulated transcript abundance, respectively, compared to the relevant controls (0 h sample is an untreated sample). Bar on the top right corner represents log 2 transformed values.

**Table 1 ijms-22-08793-t001:** *IgBBX* gene sequences and the identity of likely *A. thaliana* homologs.

Gene.	GenBank Accession No.	cDNA Length (bp)	Amino Acids Length (aa)	AtBBX Orthologs	Locus Name	*E*-Value
*IgBBX1*	MW357644	1176	312	*AtBBX6*	AT5G57660	2 × 10^−52^
*IgBBX2*	MW357645	987	328	*AtBBX4*	AT2G24790	2 × 10^−76^
*IgBBX3*	MW357646	749	197	*AtBBX19*	AT4G38960	1 × 10^−74^
*IgBBX4*	MW357647	919	192	*AtBBX22*	AT1G78600	3 × 10^−76^
*IgBBX5*	MW357648	894	246	*AtBBX24*	AT1G06040	2 × 10^−86^
*IgBBX6*	MW357649	1478	475	*AtBBX12*	AT2G33500	1 × 10^−63^

## Data Availability

The data supporting the results of this study can be obtained in the Appendix A of this article and can be obtained from the corresponding author upon reasonable request.

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
