# Peer review of "Genome-Wide Identification and Expression Analysis of BBX Transcription Factors in Iris germanica L."

_ijms, 2021, doi:10.3390/ijms22168793_

Round 1

Reviewer 1 Report

The manuscript by Wang et al presents a novel study of B-Box (BBX) genes in Iris germanica and represent a systematic approach to characterize the gene family and transcriptional regulation in plant development and stress response. Iris germanica is a popular ornamental species, thus the study provides a new information beyond widely used model species.

BBX genes are known to be involved in abiotic stress response and hormone signaling in other species. The authors analysed gene structure, phylogenetic relationships, and expression profilies of six IgBBX genes in response to hormone and stress treatments. They also confirm nuclear localization and transcription activation of IgBBX proteins. 

The manuscript is well written and structured, the figures provide sufficient information.

However, qRT-PCR analysis rise a question of data normalization: the authors only one reference gene for data analysis, whereas the MIQE guidelines suggest the use of at least two reference genes. I suggest addition of two genes for data normalization. 

Minor comments:

  1. Figure 2 legend: description of mRFP-NLS is missing.
  2. Page 4, line 101: IgBBX6 construct is missing in the text
  3. Figures 4, 5, 6: gene names should be italicized
  4. Page 6, line 119: "a 3 and 1 h exposure..." should be "a 3 and 6 h exposure"?
  5. Figure 4 and 5 legends: A, B, C should be the same as in the figures, i.e. a, b, c.
  6. Page 9, line 225: Scince should be since.

Author Response

Dear Editors and Reviewers,

Thank you and the reviewers for instructive comments on our manuscript entitled “Genome-Wide Identification and Transcriptional Expression Profiling of BBX Transcription Factors in Iris Germanica L.”. The comments are valuable and helpful for improving our paper. We have revised the manuscript carefully. The revisions in response to comments were highlighted in red letters in the text, and the detailed revisions are listed below point by point in blue letters.

Thanks for your attentions, and we are looking forward to your positive response.

Best regards,

Haiyan Yuan

Institute of Botany, Jiangsu Province and Chinese Academy of Sciences

Nanjing 210014, China

Tel: +86-25-84347086

Response to the Reviewer1’s comments

The manuscript by Wang et al presents a novel study of B-Box (BBX) genes in Iris germanica and represent a systematic approach to characterize the gene family and transcriptional regulation in plant development and stress response. Iris germanica is a popular ornamental species, thus the study provides a new information beyond widely used model species.

BBX genes are known to be involved in abiotic stress response and hormone signaling in other species. The authors analysed gene structure, phylogenetic relationships, and expression profilies of six IgBBX genes in response to hormone and stress treatments. They also confirm nuclear localization and transcription activation of IgBBX proteins.

The manuscript is well written and structured, the figures provide sufficient information.

However, qRT-PCR analysis rise a question of data normalization: the authors only one reference gene for data analysis, whereas the MIQE guidelines suggest the use of at least two reference genes. I suggest addition of two genes for data normalization.

Response: Thanks for the comment. The reference gene UBC9 has been proved to be the most suitable internal reference gene in Iris germanica in our previous studies (Wang et al., 2020). Besides, one reference gene was used as an internal control for qRT-PCR normalization, which following the reference of Lu et al., 2020 (J Exp Bot. 71, 4057-4068), Zhang et al., 2021 (Plant Cell. koab152) and Yang et al., 2021 (Int. J. Mol. Sci. 22, 7708). Therefore, in this study, we selected UBC9 as the reference gene.

Point 1: Figure 2 legend: description of mRFP-NLS is missing.

Response: Thank you for your suggestion, we have added it. (Page 4, line 103)

Point 2: Page 4, line 101: IgBBX6 construct is missing in the text.

Response: Thank you for your suggestion, we have added it. (Page 4, line 108-109)

Point 3: Figures 4, 5, 6: gene names should be italicized.

Response: Thank you for your suggestion, we have revised them. (Page 6, line 120; Page 7, line 136; Page 8, line 160)

Point 4: Page 6, line 119: "a 3 and 1 h exposure..." should be "a 3 and 6 h exposure"?

Response: Thank you for your suggestion, we have revised it. (Page 6, line 126)

Point 5: Figure 5 and 6 legends: A, B, C should be the same as in the figures, i.e. a, b, c.

Response: Thank you for your suggestion, we have revised them. (Page 7, Line 137-138; Page 8, Line 161-162)

Point 6: Page 9, line 225: Scince should be since.

Response: Thank you for your suggestion, we have revised it. (Page 9, line 233)

References

Wang, Y.J.; Zhang, Y.X.; Liu, Q.Q.; Liu, L.Q.; Huang, S.Z.; Yuan H.Y. Reference gene selection for qRT-PCR normalization in Iris germanica L.. Phyton-Int J Exp Bot. 2020, 90, 277-290.

Lu, J.; Sun, J.J.; Jiang, A.Q.; Bai, M.J.; Fan, C.G.; Liu, J.Y.; Ning, G.G.; Wang, C.Q. Alternative expressions of RcCOL4 in short day and RcCO in long day facilitate day-neutral response in Rosa chinensis. J Exp Bot. 2020, 71, 4057-4068.

Zhang, Y.; Wu, Z.C.; Feng, M.; Chen, J.W.; Qin, M.Z.; Wang, W.R.; Bao, Y.; Xu, Q.; Ye, Y.; Ma,C.; et al. The circadian-controlled PIF8-BBX28 module regulates petal senescence in rose flowers by governing mitochondrial ROS homeostasis at night. Plant Cell 2021, koab152.

Yang, L.; Cao, H.H.; Zhang, X.P.; Gui, L.X.; Chen, Q.; Qian, G.; Xiao, J.X.; Li, Z.G. Genome-wide identification and expression analysis of tomato ADK Gene family during development and stress. Int. J. Mol. Sci. 2021, 22, 7708.

Reviewer 2 Report

This paper describes the identification and characterization of BBX transcription factors in Iris Germanica. Six BBX transcription factors were cloned based on transcriptome data, and their nuclear localization, transcriptional activation ability, and responsiveness to plant hormones and stresses were analyzed.

Nuclear localization, transcriptional activation ability, and responsiveness to phytohormones and stress suggest a function of BBX transcription factors in stress response. However, the physiological function of the BBX transcription factor is not clear, and the paper is not novel enough. Even if it is challenging to take a genetic approach in Iris Germanica, complementation experiments of Arabidopsis thaliana bbx mutant and overexpression in different species can reveal the physiological function of BBX transcription factors.

In addition, the information on the transcriptome analysis used to obtain the sequence information of the IgBBX genes is not shown in the paper. If this has been reported in the past, the appropriate literature should be cited.

Author Response

Dear Editors and Reviewers,

Thank you and the reviewers for instructive comments on our manuscript entitled “Genome-Wide Identification and Transcriptional Expression Profiling of BBX Transcription Factors in Iris Germanica L.”. The comments are valuable and helpful for improving our paper. We have revised the manuscript carefully. The revisions in response to comments were highlighted in red letters in the text, and the detailed revisions are listed below point by point in blue letters.

Thanks for your attentions, and we are looking forward to your positive response.

Best regards,

Haiyan Yuan

Institute of Botany, Jiangsu Province and Chinese Academy of Sciences

Nanjing 210014, China

Tel: +86-25-84347086

Response to the Reviewer2’s comments

This paper describes the identification and characterization of BBX transcription factors in Iris Germanica. Six BBX transcription factors were cloned based on transcriptome data, and their nuclear localization, transcriptional activation ability, and responsiveness to plant hormones and stresses were analyzed.

Point 1: Nuclear localization, transcriptional activation ability, and responsiveness to phytohormones and stress suggest a function of BBX transcription factors in stress response. However, the physiological function of the BBX transcription factor is not clear, and the paper is not novel enough. Even if it is challenging to take a genetic approach in Iris Germanica, complementation experiments of Arabidopsis thaliana bbx mutant and overexpression in different species can reveal the physiological function of BBX transcription factors.

Response: Thank you very much for your instructive suggestion. Our study provided the first genome-wide analysis of the BBX gene family in Iris germanica. The stress response patterns of six IgBBX were detected under phytohormone treatments and abiotic stresses, which following the reference of Wei et al., 2020 (BMC Plant Biol. 20, 72), Haider et al., 2021 (Int. J. Mol. Sci. 22, 6585) and Tang et al., 2021 (Int. J. Mol. Sci. 10, 882). It would be very meaningful to investigate the experiments of Arabidopsis thaliana bbx mutant and overexpression in different species to reveal the physiological function of BBX transcription factors, which would be our ongoing research.

Point 2: In addition, the information on the transcriptome analysis used to obtain the sequence information of the IgBBX genes is not shown in the paper. If this has been reported in the past, the appropriate literature should be cited.

Response: Thank you for your suggestion, we added them. (Page 10, line 260-262; Page 15, line 440-442)

References

Wei, H.R.; Wang, P.P.; Chen, J.Q.; Li, C.J.; Wang, Y.Z.; Yuan, Y.B.; Fang, J.G.; Leng, X.P. Genome-wide identification and analysis of B-BOX gene family in grapevine reveal its potential functions in berry development. BMC Plant Biol. 2020, 20, 72.

Haider, M.S.; Britto, S.D.; Nagaraj, G.; Gurulingaiah, B.; Jogaiah, S. Genome-wide identification, diversification, and expression analysis of lectin receptor-like kinase (LecRLK) gene family in cucumber under biotic stress. Int. J. Mol. Sci. 2021, 22, 6585.

Tang, Y.M.; Lu, F.Z.; Feng, W.Q.; Liu, Y.; Cao, Y.; Li, W.C.; Fu, F.L.; Yu, H.Q. Genome-wide identification and expression analyses of AnSnRK2 Gene family under osmotic stress in Ammopiptanthus nanus. Int. J. Mol. Sci. 2021, 10, 882.

Reviewer 3 Report

Manuscript ID ijsm-1314176.

The m.s. entitled “Transcriptome-Wide Identification and Transcriptional Expression Profiling of BBX Transcription Factors in Iris Germanica L..” report the identification of BBX transcription factor in Iris Germanica L. and their expression analysis.

Although the data reported added new information about the characterization of BBX TF family in Iris Germanica L. ,  the manuscript present many remarks.

Major comment:

In the title, "The transcriptome wide identification ..." is misleading because the author do not describe how transcriptomics data, reported in ref. 42, were analysed to identify the BBX genes.

Which database were used? Which software were used to search the BBX domain? Also, ref 42 report roots transcriptome of I. lacteal var chinensis  in response to salts stress conditions,  that mean the library are representative of tissues specific genes and salt stress induced.

In paragraph  2.1. Identification and Phylogenetic Analysis of IgBBX Genes in I. germanica ,  the authors do not report how the IgBBX  genes were identified. Moreover in the section 4. Material and Methods paragraph  4.2. Isolation and Sequencing of Full-Length IgBBX cDNAs, are reported the full-length isolation of the IgBBX genes by using a PCR amplification but there are no other information about their identification.  In the test they only reported “Based on the transcriptome database of Iris lactea var. chinensis [42],  primer pairs (Table S1)”……  furthermore, table S1 is not present.

In the expression analysis presented in Figure 4, 5 and 6 the authors report, in the captions of the figures, the expression profiles of the IgBBX  “compared to the relevant controls” Is not clear which are the relevant controls.

Minor comment:

Line  46…. CO regulates the expression of FT, a prominent floral inducer. Please indicate Flowering Locus T (FT) gene,  …

Pag. 5 - Figure 3……. Please Indicate the date as show “in left side and in right side”.

Pag. 5….        2.3. Transcription Profiling of IgBBX Genes    please change in “Expression profile of IgBBX in different plant tissues.

Pag. 6…     2.4. Transcription Profiling of IgBBX Genes after Treatment with Phytohormones  change in  “Expression profile of IgBBX genes after ABA, MeJA and SA Phytohormones treatment”

Pag.11 - line 305   “UBC9 gene was employed as a reference sequence.” Please indicate the accession number and if it belongs to I.germanica L.

There are others minor comment but all in all the manuscript should be improved in all sections by adding more details on the identification and characterization of BBX genes of Iris germanica L.

Author Response

Dear Editors and Reviewers,

Thank you and the reviewers for instructive comments on our manuscript entitled “Genome-Wide Identification and Transcriptional Expression Profiling of BBX Transcription Factors in Iris Germanica L.”. The comments are valuable and helpful for improving our paper. We have revised the manuscript carefully. The revisions in response to comments were highlighted in red letters in the text, and the detailed revisions are listed below point by point in blue letters.

Thanks for your attentions, and we are looking forward to your positive response.

Best regards,

Haiyan Yuan

Institute of Botany, Jiangsu Province and Chinese Academy of Sciences

Nanjing 210014, China

Tel: +86-25-84347086

Response to the Reviewer3’s comments

The m.s. entitled “Transcriptome-Wide Identification and Transcriptional Expression Profiling of BBX Transcription Factors in Iris Germanica L.”. report the identification of BBX transcription factor in Iris Germanica L. and their expression analysis.

Although the data reported added new information about the characterization of BBX TF family in Iris Germanica L., the manuscript present many remarks.

Major comment:

Point 1: In the title, "The transcriptome wide identification ..." is misleading because the author do not describe how transcriptomics data, reported in ref. 42, were analysed to identify the BBX genes.

Response: Thank you for your suggestion, we have revised it. (Page 1, Line 2)

Point 2: Which database were used? Which software were used to search the BBX domain?

Response: Thank you for your suggestion, we have added them. (Page 10, line 260-262; Page 10, line 271-273; Page 15, line 440-442)

Point 3: Also, ref 42 report roots transcriptome of I. lacteal var. chinensis in response to salts stress conditions, that mean the library are representative of tissues specific genes and salt stress induced.

Response: Thanks for the comments. The transcriptome of I. lacteal var. chinensis is in response of root to salts stress conditions. We only used this transcriptome database to obtain the BBX sequences for gene cloning.

Point 4: In paragraph 2.1. Identification and Phylogenetic Analysis of IgBBX Genes in I. germanica, the authors do not report how the IgBBX genes were identified.

Response: Thank you for your suggestion, we have added them. (Page 2, line 73-79; Page 17, line 476-481)

Point 5: Moreover in the section 4. Material and Methods paragraph 4.2. Isolation and Sequencing of Full-Length IgBBX cDNAs, are reported the full-length isolation of the IgBBX genes by using a PCR amplification but there are no other information about their identification.  In the test they only reported “Based on the transcriptome database of Iris lactea var. chinensis [42], primer pairs (Table S1) furthermore, table S1 is not present.

Response: Thank you for your suggestion, we have added them. (Page 10, line 271-273; Page 17, line 482)

Point 6: In the expression analysis presented in Figure 4, 5 and 6 the authors report, in the captions of the figures, the expression profiles of the IgBBX “compared to the relevant controls” Is not clear which are the relevant controls.

Response: Thanks for the comments.

In Figure 4, the expression of IgBBX in roots was used as the relevant controls. In Figure 5 and 6, the expression of IgBBX at 0 h was used as the relevant controls. Blue and red indicate down-regulated and up-regulated transcript abundance, respectively.

Minor comment:

Point 1: Line 4. CO regulates the expression of FT, a prominent floral inducer. Please indicate Flowering Locus T (FT) gene.

Response: Thank you for your suggestion, we have added it. (Page 2, line 46)

Point 2: Pag. 5. Figure 3 please indicate the date as show “in left side and in right side”.

Response: Thank you for your suggestion, we have added them. (Page 5, Line 111-112)

Point 3: Pag. 5. 2.3. Transcription Profiling of IgBBX Genes please change in “Expression profile of IgBBX in different plant tissues.

Response: Thank you for your suggestion, we have revised it. (Page 5, Line 114)

Point 4: Pag. 6. 2.4. Transcription Profiling of IgBBX Genes after Treatment with Phytohormones change in “Expression profile of IgBBX genes after ABA, MeJA and SA Phytohormones treatment”.

Response: Thank you for your suggestion, we have revised it. (Page 6, Line 124)

Point 5: Pag.11 line 305 “UBC9 gene was employed as a reference sequence.” Please indicate the accession number and if it belongs to I. germanica L.

Response: Thank you for your suggestion, we have revised it. (Page 11, line 318)

Point 6: There are others minor comment but all in all the manuscript should be improved in all sections by adding more details on the identification and characterization of BBX genes of Iris germanica L.

Response: Thanks for your suggestion. We have improved the English of the manuscript by native English speakers and the editing certificate is as follows. The detail revisions of English are highlighted in red letters. (Line 12, Line 15, line 17, line 35-37, Line 41, Line 56, Line 87, Line 93-94, Line 98, Line 104, Line 106, Line 126, Line 134-135, Line 142-143, Line 159, Line 166, Line 180-181, Line 185, Line 187, Line 190-191, Line 194, Line 204, Line 212, Line 220-221, Line 224-225, Line 243-244, Line 285, Line 291, Line 294, Line 303, Line 305, Line 319, Line 327).

Round 2

Reviewer 2 Report

The reviewer understood the novelty and importance of this paper by the author's response. In addition, the paper has been improved by other reviewer's suggestions. Therefore, the reviewer supports the acceptance of this paper.

Author Response

Dear Editors and Reviewers,

Thank you and the reviewers for instructive comments on our manuscript entitled “Genome-Wide Identification and Expression Analysis of BBX Transcription Factors in Iris Germanica L.”. The comments are valuable and helpful for improving our paper. We have revised the manuscript carefully. The revisions in response to comments were highlighted in red letters in the text, and the detailed revisions are listed below point by point in blue letters.

Thanks for your attentions, and we are looking forward to your positive response.

Best regards,

Haiyan Yuan

Institute of Botany, Jiangsu Province and Chinese Academy of Sciences

Nanjing 210014, China

Tel: +86-25-84347086

Response to the Reviewer3’s comments

In the present version the authors have significantly improved the data reported and the English editing of the manuscript with a consequently a greater understanding by readers of the results reported.

Although the authors have changed the title of the manuscript, I suggest to change as: Genome-Wide Identification and Expression Analysis of BBX Transcription Factors in Iris Germanica L.

Response: Thank you for your suggestion, we have revised it. (Page 1, Line 2)

Minor comment

Point 1: Line 15-16 A transient expression experiment revealed that all six BBX proteins were deposited in the nucleus. Please change in “A transient expression analysis revealed that all six BBX proteins were localized in the nucleus.”

Response: Thank you for your suggestion, we have revised it. (Page 1, Line 14-15)

Point 2: Line 75-76 The conserved domains of IgBBX were identified by the SMART and Pfam database. Please indicate the website and reports in the bibliographic list.

Response: Thank you for your suggestion, we have added them. (Page 10, Line 273-275)

Point 3: Line 125 compared to the relevant controls. Please indicate which is the relevant control (roots, leaf, flower?).

Response: Thank you for your suggestion, we have added it. (Page 6, Line 122)

Point 4: Line 144 compared to the relevant controls. If the relevant controls correspond to the 0 h sample, please indicate it as an untreated sample. Review all the legends of genes expression.

Response: Thank you for your suggestion, we have added them. (Page 7, Line 139; Page 8, Line 163; Page 17, Line 474, Line 478)

Reviewer 3 Report

[IJMS] Manuscript ID: 1314176

In the present version the authors have significantly improved the data reported and the English editing of the manuscript with a  consequently  a greater understanding by readers of the results reported.

Although the authors have changed the title of the manuscript, I suggest to change as: Genome-Wide Identification and  Expression Analysis of BBX Transcription Factors in Iris Germanica L.

Minor comment

Line 15-16 … A transient expression experiment revealed that all six BBX proteins were deposited in the nucleus. Please change in  “  A transient expression analysis revealed that all six BBX  proteins were localized in the nucleus.”

Line 75-76 …. The conserved domains of Ig BBX were identified by the SMART and Pfam database. Please indicate the website and reports in the bibliographic list.

Line 125 …. compared to the relevant controls.  Please indicate which is the relevant control ( roots, leaf, flower ?)

Line 144… compared to the relevant controls.  If the relevant controls correspond to the 0h sample, please indicate it as an untreated sample.  Review all the legends of genes expression.

Author Response

(The authors gave the same response as above.)
